

# Invasion history of *Harmonia axyridis* (Pallas, 1773) (Coleoptera: Coccinellidae) in Ecuador

Diego F. Cisneros-Heredia[1,2] and Emilia Peñaherrera-Romero[1,2]

[1] Colegio de Ciencias Biológicas y Ambientales COCIBA, Universidad San Francisco de Quito USFQ, Quito, Ecuador
[2] Instituto de Diversidad Biológica Tropical iBIOTROP, Museo de Zoología & Laboratorio de Zoología Terrestre, Universidad San Francisco de Quito USFQ, Quito, Ecuador

## ABSTRACT

*Harmonia axyridis* is a ladybird extensively used around the world for biological control of agricultural pests. However, it has become invasive in several countries, producing negative ecological and socio-economic impacts. Herein, we review the invasion history of the Harlequin Ladybird *Harmonia axyridis* (Pallas, 1773) in Ecuador. Although first reported in Ecuador in 2012, museum specimens date back to 2004 and it is currently established across the country, especially along the Andean region. Due to its invasive nature, further studies are urgently needed to evaluate possible impacts of *H. axyridis* on the Ecuadorian biodiversity and agroindustry.

## INTRODUCTION

Numerous species have arrived in regions they would have never reached on their own due to human-mediated processes (*Ricciardi, 2007*; *Boivin et al., 2016*). Although many non-native species are unable to thrive in new environments, some are successful and become invasive by establishing fast-growing, spreading populations. Invasive species have been described as major drivers of current biodiversity changes due to their contribution to biota homogenization, alteration of biological communities and ecosystem functions, and socio-economic impacts on humans (*Chapin et al., 2000*; *Daszak, Cunningham & Hyatt, 2000*; *Crooks, 2002*; *O'Dowd, Green & Lake, 2003*; *Clavero & García-Berthou, 2005*; *Mace et al., 2005*; *Doody et al., 2009*; *Pejchar & Mooney, 2009*; *Ricciardi et al., 2013*; *Simberloff et al., 2013*; *Bellard, Cassey & Blackburn, 2016*; *Doherty et al., 2016*; *Cisneros-Heredia, 2018*).

*Harmonia axyridis* (*Pallas, 1773*), commonly referred to as Harlequin Ladybird or Asian Multicolored Ladybeetle, is a member of the family Coccinellidae native to East Asia (*Orlova-Bienkowskaja, Ukrainsky & Brown, 2015*). It has been deliberately translocated as a control agent in America, Europe and Africa since the beginning of the 20th century, establishing naturalized and expanding populations in the three continents, becoming invasive (*Lombaert et al., 2010*; *Brown et al., 2011*). It is a successful invader due to its wide dietary range, ability to establish and disperse, and robustness and

Corresponding author
Diego F. Cisneros-Heredia,
diego.cisnerosheredia@gmail.com

flexibility of its immune system (*Roy, Brown & Majerus, 2006*; *Vilcinskas, Mukherjee & Vogel, 2013*). *Harmonia axyridis* is considered to be the most invasive ladybird on Earth (*Roy, Brown & Majerus, 2006*).

*Harmonia axyridis* is a voracious predator of agricultural pests, consuming soft-bodied Sternorrhyncha hemipterans as essential preys, that is aphids, coccids, psyllids and adelgids (*Roy, Brown & Majerus, 2006*). However, *H. axyridis* has a wider dietary range and is an interguild and intraguild polyphagous predator, being able to consume immature stages of coccinellids and other coleopterans, lepidopterans, neuropterans, dipterans, tetranychid mites and plant material such as fruits, pollen, nectar, leaves and seeds (*Koch, 2003*; *Koch et al., 2004*; *Berkvens et al., 2008*; *Koch & Galvan, 2008*; *Moser, Harwood & Obrycki, 2008*; *Roy & Wajnberg, 2008*; *Martins et al., 2009*; *Lucas, 2012*; *Michaud, 2012*). In general, *H. axyridis* is considered one of the top predators in aphidophagous and coccidophagous guilds, largely free from predation pressure, and regulated more by bottom-up than top-down forces (*Lucas, 2012*). *Harmonia axyridis* seems to dominate confrontations with other coccinellid species, exerting strong intraguild predation pressure (*Pell et al., 2008*; *Ware & Majerus, 2008*; *Lucas, 2012*; *Katsanis et al., 2013*). Due to its polyphagy and guild interactions, non-native populations of *H. axyridis* have adverse effects on native biodiversity and agroindustry by attacking non-target arthropods, modifying the structure and dynamics of invertebrate assemblages, replacing or marginalising native coccinellids by competition and predation, and feeding on commercial fruits or damaging agricultural products (*Koch, 2003*; *Koch et al., 2004*; *Koch & Galvan, 2008*; *Honěk, 2012*; *Lucas, 2012*).

In America, the first translocation of *Harmonia axyridis* was to the USA in 1916, and recurrent introductions to USA and Canada occurred between 1964 and 1983 (*Gordon, 1985*; *Hoebeke & Wheeler, 1996*). The first established feral populations in North America were recorded in 1988 in eastern USA (*Chapin & Brou, 1991*), in 1991 in western USA (*LaMana & Miller, 1996*), and in 1994 in Canada (*Coderre, Lucas & Gagné, 1995*). The two USA populations originated from independent introductions from the species' native range (*Lombaert et al., 2010*), and the Canadian population apparently spread from eastern USA (*McCorquodale, 1998*). All subsequent successful introductions of *H. axyridis* across America have seemingly sourced from eastern USA (*Lombaert et al., 2010*). Mexican populations descend from eastern USA stocks deliberately released in northern Mexico (ca. 1997) and southeastern Mexico (1999–early 2000s) (*Quiñonez Pando & Tarango Rivero, 2005*; *Barrera & López-Arroyo, 2007*). First translocations to Argentina (1986) and Chile (1998) used parental stocks from France but were unsuccessful in establishing populations (*García, Becerra & Reising, 1999*; *Saini, 2004*; *Grez et al., 2010*). Naturalised populations reported in Argentina in 2001, southern Brazil in 2002 and Chile in 2003 descend from at least two different eastern USA stocks (*Almeida & Silva, 2002*; *Saini, 2004*; *Grez et al., 2010*; *Lombaert et al., 2010*; *Brown et al., 2011*).

The oldest known naturalised populations of *H. axyridis* in South America were established in Colombia, where it was first collected in 1989 (*Kondo & González, 2013*). Since *H. axyridis* was extensively raised and shipped in the USA in the 1980s and 1990s (*Tedders & Schaefer, 1994*), and based on available dates, Colombian populations may also

descend from eastern USA stock. It is probable that unrecorded international shipments were sent to Colombia, Argentina, and Chile—possibly to private farmers, thus the absence of public records. Subsequent South American records come from Ecuador (2004, see below), Paraguay (2006, *Silvie et al., 2007*), Uruguay (2007, *Nedvěd & Krejčík, 2010*; *Serra, González & Greco-Spíngola, 2013*), Peru (ca. 2010, *Grez et al., 2010*), and Venezuela (ca. 2014, *Solano & Arcaya, 2014*). It has not been formally reported from Guyana, Suriname and Bolivia (*Camacho-Cervantes, Ortega-Iturriaga & Del-Val, 2017*; *Hiller & Haelewaters, 2019*), but a recent citizen-science record shows that it is already present in Bolivia (*Maslowski, 2020*). Reports of *H. axyridis* from Central America have only recently been published, but the oldest records date back to 1988 and 1996—from Costa Rica. The species is currently established in most Central American countries, but is has not been reported from Belize, El Salvador and Nicaragua (*Hiller & Haelewaters, 2019*).

*Harmonia axyridis* was first recorded in Ecuador in 2012 by *González & Kondo (2012)* who reported 11 specimens collected in 2012 in deciduous forests on La Ceiba and Laipuna natural reserves (762 and 828 m elevation, respectively), province of Loja, in the extreme southwestern lowlands of Ecuador. *Cornejo & González (2015)* reported the species from mangroves on Santay Island (at sea level), province of Guayas, southwestern Ecuador. *González (2015)* reported *H. axyridis* from the provinces of Azuay, Guayas and Loja, but without referencing any voucher specimen from Azuay. *Guamán Montaño (2017)* presented photographs of *H. axyridis* from El Pangui (830 m elevation), province of Zamora-Chinchipe, providing the first reports on the south-eastern slopes of the Andes of Ecuador. Geographic and ecological data of *H. axyridis* in Ecuador are scarce. Herein, we discuss the distribution, natural history, and introduction history of *Harmonia axyridis* in Ecuador, showing that it has been present at least since 2004 and is currently widespread across the country.

## MATERIALS AND METHODS

Coccinellid beetles were opportunistically collected since 2015 during field surveys of the Universidad San Francisco de Quito USFQ at 17 localities across northern Ecuador, (Table S1; Figs. 1 and 2). Field surveys were conducted by the authors, usually with 8–15 undergraduate students of the USFQ Biology program. All specimens were found by searching vegetation to look for adults and larvae. Collected specimens were euthanized by immersion in 70% ethanol or by placing in a killing jar and stored. An Olympus Research Stereomicroscope System SZX16 outfitted with an Olympus DP73 digital colour camera was used to examine specimens. Voucher specimens collected during our surveys are deposited at the Museo de Zoología (ZSFQ), Universidad San Francisco de Quito USFQ, Ecuador. Research permits were issued by Ministerio de Ambiente del Ecuador, 001-16IC-FLO-FAU-DNB/MA, 018-2017-IC-FAU-DNB/MAE, 019-2018-IC-FAU-DNB/MAE and 006-2015-FAU-DPAP-MA.

We reviewed the entomological collections of Museo de Zoología, Pontificia Universidad Católica del Ecuador, Quito (QCAZ), and Instituto Nacional de Biodiversidad INABIO, Quito (MECN). Published information on *Harmonia axyridis* in Ecuador was synthesised based on a literature review using the library systems of King's College London

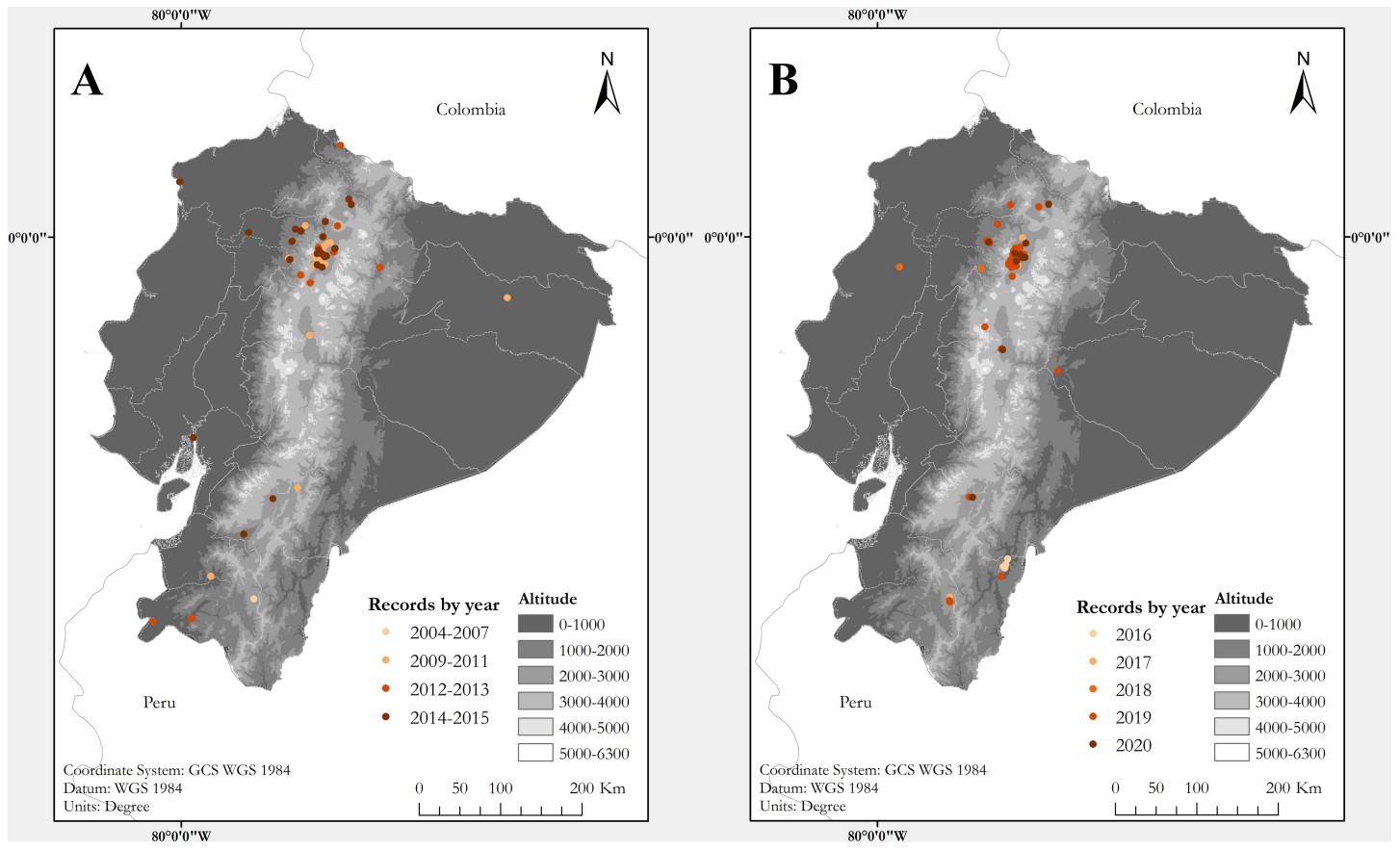

**Figure 1** **Maps of Ecuador showing known localities of the Harlequin Ladybird *Harmonia axyridis* (Pallas, 1773) by year.** (A) Map showing records between 2004 and 2015. (B) Records between 2016 and 2020. Each locality point may correspond to several records (see Table S1)

and Universidad San Francisco de Quito USFQ and Google Scholar™ scholarly text search (https://scholar.google.com). Relevant references were gathered using the search terms 'Coccinellidae,' '*Harmonia*,' '*Harmonia axyridis*,' 'Mariquita,' 'Ladybird,' 'Ladybug' and 'Lady beetle', each one combined with 'Ecuador' by the Boolean operator 'AND'. Since *H. axyridis* has a diagnostic colouration pattern that allows its identification in photographs, we assembled data from photographic vouchers using the search engines of Flickr™ (https://www.flickr.com, by Yahoo!) and iNaturalist.org™ (http://www.inaturalist.org, by California Academy of Sciences and the National Geographic Society) through GeoCat (*Bachman et al., 2011*; http://geocat.kew.org/) using the same search terms used for text searches. All searches were run on 10 February 2019 using on-site search engines and were not limited by study type, study design, or language. iNaturalist searches were rerun on 09 August 2019 and on 01 April 2020.

All localities, based on field surveys, literature, museum and photographic records, were georeferenced manually in Google Earth™ mapping service (7.3.1.5491 release by Google, Inc. on July 2018) based on direct information (coordinates and altitudinal data) when available, and additional data relevant to obtain an accurate and precise positioning, including catalogue and field notes, following recommendations by *Wieczorek, Guo &*

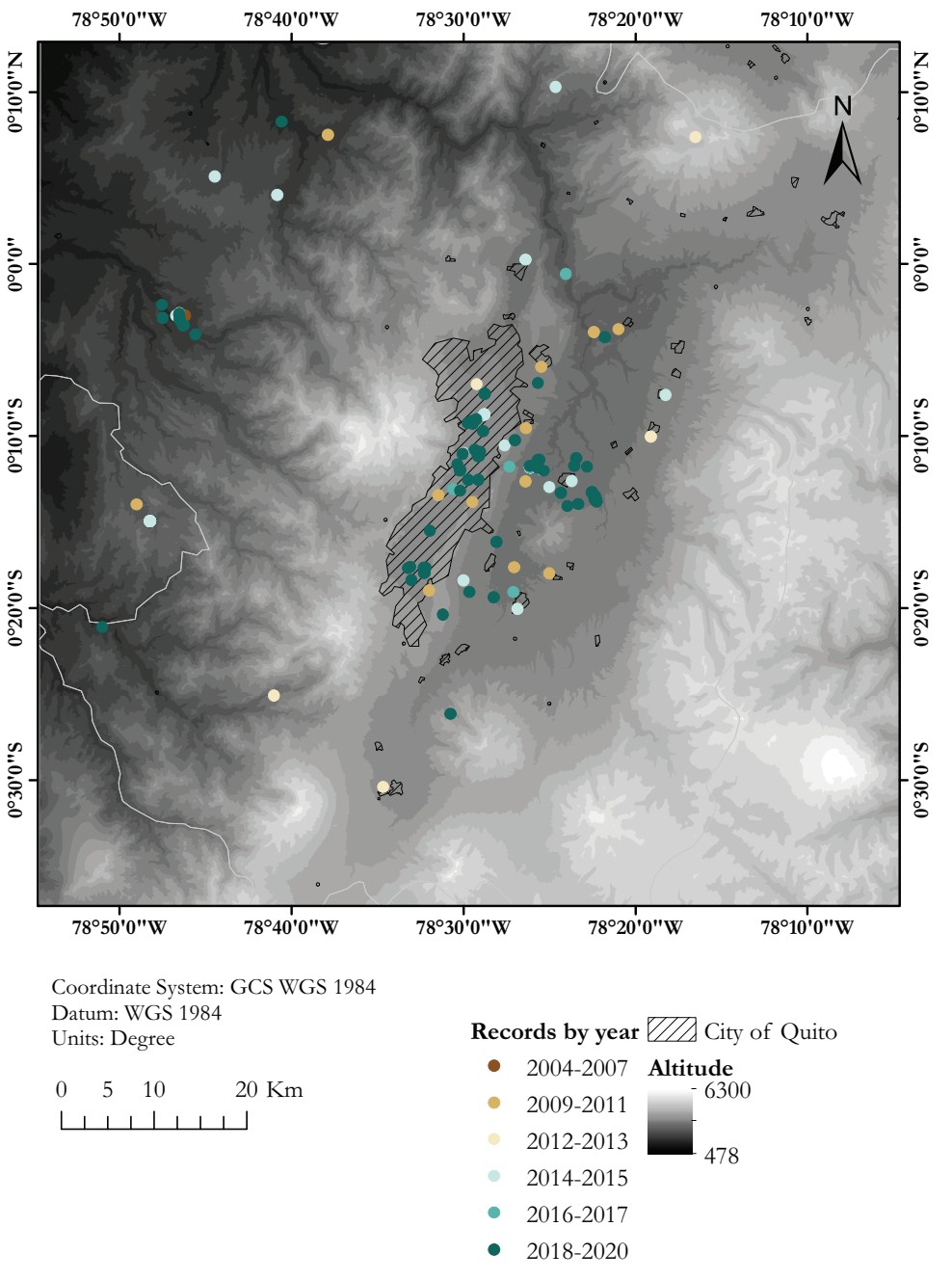

**Figure 2 Map showing known localities of the Harlequin Ladybird in the valley of Quito, capital city of Ecuador, by year.** Dashed area marks the city of Quito. Each locality point may correspond to several records (see Table S1)

*Hijmans (2004).* All localities were reviewed and validated individually, and coordinates were amended when incorrectly georeferenced in the source (Table S1). We determined the position most closely related with the locality description using toponymic information based on the Geographic Names Database, containing official standard names approved by the United States Board on Geographic Names and

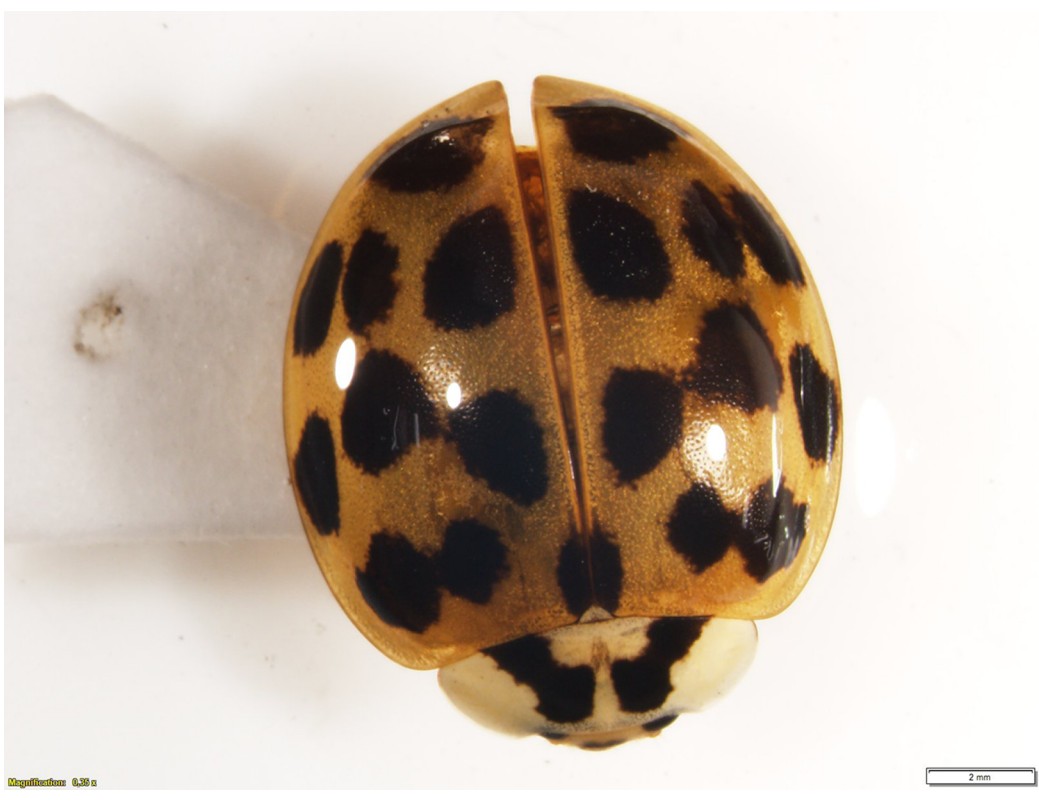

**Figure 3 *Harmonia axyridis* from Ecuador.** Photo of specimen ZSFQ-I058 from Cumbayá (USFQ campus), province of Pichincha, Ecuador, showing the typical habitus of Ecuadorian populations.

maintained by the National Geospatial-Intelligence Agency (http://geonames.nga.mil/gns/html/), OpenMapStreet data available under the Open Database Licence (http://www.openstreetmap.org), and gazetteers for Ecuador (*Brown, 1941*; *Peters, 1955*; *Lynch & Duellman, 1997*).

## RESULTS

In total, we collected information for 294 records of *Harmonia axyridis* from 53 localities in Ecuador (Table S1; Figs. 1 and 2), including: 106 specimens collected during field surveys and deposited at ZSFQ, 11 individuals recorded during field surveys but uncollected, 37 museum specimens (37 at QCAZ, none at MECN), 118 individuals recorded in iNaturalist, five photographic records from Flickr, and 17 literature records (*González & Kondo, 2012*; *Cornejo & González, 2015*; *Guamán Montaño, 2017*). Most records presented herein correspond to adult individuals, but larvae and pupa were recorded across the Andes (Table S1). Specimens were identified as *Harmonia axyridis* by its characteristic morphology (Fig. 3), including: upper surfaces of elytra not hairy, distinct transverse fold at rear of elytra, underside of abdomen at least partially orange, brown to orange legs (*Koch, 2003*; *Roy et al., 2016*). One phenotypic colour form was found: f. *succinea*, with ground colour of pronotum white to light brown with M-shaped

black marks, ground colour of elytra bright orange, usually with nine black elytral spots (2-3-3-1) on each elytra, and a scutellary spot (*Dobzhansky, 1933*; *Tan & Li, 1934*; *Koch, 2003*; *Brown et al., 2008a*; *Roy et al., 2013*; *Roy et al., 2016*).

The first specimens of *H. axyridis* in Ecuador were collected on both geographical extremes of the country: in 2004 at Mindo, northwestern Ecuador, and in 2007 at Loja, in southwestern Ecuador (Table S1). Both localities are separated by about 450 km and now have established populations. Our study reveals that *H. axyridis* is now established in all main biogeographic regions of Ecuador, from sea level to at least 4,020 m elevation, across 16 provinces (administrative geopolitical divisions of Ecuador) (Table S1; Fig. 1). At least one record (at Estación Científica Yasuní) may correspond to a hitchhiking individual, since no established population in the Amazonian lowlands has been confirmed. The only regions where we did not find records are the southern Amazonian lowlands and the Galapagos Archipelago.

Almost two-thirds of the localities where we found records of *axyridis* are anthropogenic habitats (51% are urban green spaces in mid-size towns and large cities, and 20% are agricultural lands; Table S1). Most of the records of *H. axyridis* in Ecuador come from the Andes, a region with significant agricultural and urban areas, including the capital city, Quito, and its metropolitan district. In urban green spaces, *H. axyridis* usually occupies gardens and parks dominated by non-native plants. *Harmonia axyridis* has been found also in 12 different ecosystems with native vegetation, usually collected along road borders and near human settlements.

During our surveys, *H. axyridis* was associated with the following plants (local names and families in parentheses): *Ambrosia arborescens* (Marco, Asteraceae), *Baccharis latifolia* (Chilca, Asteraceae), *Citrus × limon* (Limonero, Rutaceae), *Chusquea* sp. (Suro, Poaceae), *Cupressus* sp. (Ciprés, Cupressaceae), *Delostoma integrifolium* (Yalomán, Bignoniaceae), *Erigeron* sp. (Asteraceae), *Ficus benjamina* (Ficus, Moraceae), *Hibiscus rosa-sinensis* (Cucarda, Malvaceae), *Lantana camara* (Tupirrosa or Supirrosa, Verbenaceae), *Ligustrum* sp. (Oleaceae), *Lilium* sp. (Lirio amarillo, Liliaceae), *Ocimum basilicum* (Albahaca, Lamiaceae), *Petroselinum crispum* (Perejil, Apiaceae), *Prunus persica* (Durazno, Rosaceae), *Prunus serotina capuli* (Capulí, Rosaceae), *Rosa* sp. (Rosa, Rosaceae), *Senna multiglandulosa* (Chinchín, Fabaceae), *Solanum nigrescens* (Yerbamora, Solanaceae), *Tecoma stans*, (Cholán, Bignoniaceae), *Trifolium repens* (Trébol blanco, Fabaceae) and *Verbesina sodiroi* (Asteraceae). The following coccinellids were found in sympatry with *H. axyridis* during our surveys at different localities: *Brachiacantha* sp. cf. *anita* (Mindo), *Cheilomenes sexmaculata* (Quito, Cumbayá, Tumbaco), *Cycloneda ecuadorica* (Guajalito, San Vicente), *C. emarginata* (Guajalito, Loja), *C. sanguinea* (Cumbayá, San Vicente), *Epilachna monovittata* (Guajalito), *E. flavofasciata* (Guajalito), *E. paenulata* (Quito, Mindo), *Hippodamia convergens* (Quito, Lumbisí, Cumbayá, Tumbaco, Yaruquí, Guajalito, San Vicente, Mindo, Loja), *Mulsantina mexicana* (Cumbayá, Yaruquí, Guajalito), *Neda norrisi* (San Antonio de Pichincha) and *Rodolia cardinalis* (San Antonio de Pichincha, Cumbayá). No parasitoids were detected.

Almost 30% of our field records of *H. axyridis* come from the Cumbayá-Tumbaco valley, an inter-Andean valley near Quito (Fig. 2), in northern Ecuador, where we had a

higher sampling effort and were able to observe the coccinellid community in more detail. We found six coccinellid species in sympatry in gardens, parks and agricultural areas of the Cumbayá-Tumbaco valley: *Harmonia axyridis*, *Cheilomenes sexmaculata*, *Hippodamia convergens*, *Mulsantina mexicana*, *Rodolia cardinalis* and *Cycloneda sanguinea*. *Harmonia axyridis* was the most common species in green urban areas, but is uncommon in agricultural areas, where *H. convergens* was dominant. *Rodolia cardinalis* and *C. sexmaculata* were found in syntopy with *H. axyridis*. By 2017, *H. convergens* was almost absent in most urban green areas of the Cumbayá-Tumbaco valley, remaining common only in agricultural areas. *Cheilomenes sexmaculata*, an apparently recent arrival—first recorded in the area on 2017, is nowadays becoming the second most common coccinellid in urban green areas of Quito–Cumbayá–Tumbaco, although still with a patchy distribution.

## DISCUSSION

Extensive and intensive entomological studies conducted in northern Ecuador up to 2001 did not record *Harmonia axyridis* (*Cardona, López-Avila & Valarezo, 2005*; *Carvajal, 2005*). Thus, the first naturalised populations of *H. axyridis* in Ecuador probably became established between 2001 and 2004, possibly as a result of intentional releases. Introduction of ladybirds has a long history in Ecuador; for example in 1978, official national authorities released 24 million individuals of *Hippodamia convergens* in the city of Quito and surroundings, in an attempt to control *Icerya purchasi* (*Molineros Andrade, 1984*). However, it is also possible that Ecuadorian populations spread from southern Colombia, since the oldest Colombian records occurred very close to the Ecuadorian border (*Kondo & González, 2013*). The presence of earliest Ecuadorian localities on opposite sides of the country (Mindo and Loja) and the absence of geographically intermediate records could suggest that Ecuadorian populations had two independent origins. However, museum records are biased due to limited collection efforts in the central provinces of Ecuador. Furthermore, spread rate of *H. axyridis* may be extremely fast and compensate for the distance between the localities (58–144.5 km/year in the UK, *Brown et al., 2008b*; 200 km/year in Slovakia, *Roy et al., 2016*; 185 km/year in Chile, *Grez et al., 2016*; 442 km/year in USA-Canada, *McCorquodale, 1998*; 500 km/year in South Africa, *Stals, 2010*). If the southern Ecuadorian population is demonstrated to have an independent origin, they may have been the source of the northern Peru populations, that remained unrecorded during extensive surveys in 2006 (*Miró-Agurto & Castillo-Carrillo, 2010*) and became established around 2010 (*Grez et al., 2010*).

The highest record of *H. axyridis* in Ecuador, at 4,020 m at Mojanda, is also the highest record worldwide, 500 m higher than the upper elevational ranges reported by *Grez et al. (2017)* and *González, Bustamante & Grez (2019)*. Unfortunately, no ecological information was associated with that specimen. Lowland records mainly come from the Pacific lowlands and western Andean slopes, but also from the Amazonian foothills and lowlands. The Andean region was predicted as suitable for the expansion of *H. axyridis* by distribution models analysed by *Koch, Venette & Hutchison (2006)* and

*Poutsma et al. (2008)*, and although they predicted the expansion of *H. axyridis* across different habitats of America, their models did not show lowland forest. Interestingly, Ecuadorian records come from a variety of habitats, including forest and shrublands, evergreen and semideciduous vegetation, and across the urban-agricultural matrix. However, the most extensive and dense populations were found in urban areas.

It is likely that *H. axyridis* will keep spreading across most of Ecuador, especially in urban and agricultural environments, having effects on the diverse Ecuadorian fauna of coccinellids due to competition, exclusion, and intraguild predation. *Harmonia axyridis* may significantly impact predatory arthropod guilds, interfering with invertebrate population dynamics, potentially producing impacts on native aphidophage groups and agricultural pests (*Lucas, Gagné & Coderre, 2002*; *Koch, 2003*; *Pervez & Omkar, 2006*; *Koch & Galvan, 2008*; *Ducatti, Ugine & Losey, 2017*). In particular, the arrival of *H. axyridis* to the Galapagos Archipelago could be problematic, due to the vulnerability of island ecosystems to impacts on endemic and native invertebrates and profound irruptions on trophic interactions (*Causton et al., 2006*).

*Harmonia axyridis* has attained the status of agricultural pest in North America because it feeds opportunistically on fruit when prey is scarce and acts as a wine contaminant (*Koch et al., 2004*; *Koch & Galvan, 2008*). Grapes, apples, peaches, plums, pears, raspberries, among other fruits, have been reported to be consumed by *H. axyridis*, blemishing the fruits and reducing their value (*Majerus, Strawson & Roy, 2006*; *Koch & Galvan, 2008*; *Guedes & Almeida, 2013*). All these fruits are produced in Ecuador, usually for local consumption but, in recent decades, have become important exportation products. Fruit crops occupy over 1600 km$^2$ in Ecuador (excluding bananas), are produced by more than 120,000 farmers, and represent a small but growing sector in non-traditional agricultural exports in the country—contributing to ca. 4.4% of all non-traditional exportations (*Viera et al., 2016*; *Verdugo-Morales & Andrade-Díaz, 2018*; *Banco Central del Ecuador (BCE), 2020*). Most Ecuadorian fruit crops are located across the highlands and western lowlands of the country (*Niegel, 1992*; *Huttel, Zebrowski & Gondard, 1999*), coinciding with areas where *H. axyridis* is expanding.

Wine contamination has been described as the most important agricultural impact of *H. axyridis*. Adults aggregate on injured grapes and can be disturbed or crushed during harvesting or pressing, releasing haemolymph that affects wine quality by causing unpleasant odour and taste—that is ladybug taint (*Pickering et al., 2004*, *2008*; *Koch & Galvan, 2008*). Although table and wine grapes have been grown in Ecuador for local consumption since the 16th century (*Popenoe, 1924*), commercial production has only been fostered in recent decades (*El Comercio, 2017*; *Revista Líderes, 2012*, *2013*; *Viera et al., 2016*). Vineyards in Ecuador have increased from 0.6 km$^2$ in 1985 to more than 2 km$^2$ today—and are expected to reach 10 km$^2$ in the near future (*Niegel, 1992*; *El Comercio, 2017*; *Revista Líderes, 2012*, *2013*). Established populations of *H. axyridis* have been reported in all areas where Ecuadorian wineries are situated (i.e. provinces of Guayas, Pichincha, Azuay and El Oro). Although Ecuadorian wine production is still modest, its presence in national and international markets is expanding (*ProEcuador, 2017*) and ladybug taint could negatively impact this growing industry.

## CONCLUSIONS

Harlequin Ladybird *Harmonia axyridis* currently holds established populations across Ecuador. It was introduced to the country at the beginning of the 21st century and, with a fast spread rate, nowadays occupy most Andean highlands (including the highest elevation worldwide at 4,020 m), and it is expanding across the Pacific and Amazonian lowlands. Information on the coccinellids of Ecuador is limited and fragmentary. It is important to increase research on the diversity, distribution, natural history, ecology and socio-economic effects of coccinellids in the country. Information is needed across urban-agricultural-natural matrices, in order to evaluate the impacts of *H. axyridis* and other non-native species. Research on the impacts of *H. axyridis* in Ecuadorian agribusiness, especially fruit and wine production, is urgently needed.

## ACKNOWLEDGEMENTS

We express our gratitude to Ana Nicole Acosta-Vásconez, Mateo Dávila-Játiva and Izan Chalen for their assistance, and to the students of the USFQ courses of Introduction to Biology and Zoology (years 2015, 2016, 2017, 2018, 2019) for their help in finding some of the populations of *Harmonia axyridis* herein reported. We thank the following people for provision of support and working space in their respective institutions or for the loan of specimens under their care: Santiago Villamarín (INABIO), Alvaro Barragán and Clifford Keil (QCAZ), and Giovani Ramón (ZSFQ). We are grateful to Carlos Ruales for helping us to find some key literature about the first introductions of ladybirds in Ecuador and to all citizens scientist that continuously contribute to iNaturalist. We thank Peter Brown and Lucia Almeida for their comments on a previous version of this article.

### Funding

This study was supported by Universidad San Francisco de Quito USFQ (Research Funds for projects ID 35 "Biodiversity of urban and rural areas of Ecuador", and ID 1057 "Impacts of habitat changes on the biological diversity of the northern tropical Andes", Outreach project "Celebrando la Naturaleza" 2017–2020, and Publication Fund to Diego F. Cisneros-Heredia) and operative funds assigned to Instituto de Diversidad Biológica Tropical iBIOTROP, Museo de Zoología & Laboratorio de Zoología Terrestre, Colegio de Ciencias Biológicas y Ambientales COCIBA and by Programa "Becas de Excelencia" of Secretaría de Educación Superior, Ciencia, Tecnología e Innovación SENESCYT, Ecuador. The funders had no role in study design, data collection and analysis, decision to publish, or preparation of the manuscript.

### Grant Disclosures

The following grant information was disclosed by the authors:
Universidad San Francisco de Quito USFQ.

Instituto de Diversidad Biológica Tropical iBIOTROP, Museo de Zoología & Laboratorio de Zoología Terrestre.

## Competing Interests

The authors declare that they have no competing interests.

## Author Contributions

- Diego F. Cisneros-Heredia conceived and designed the experiments, performed the experiments, analyzed the data, prepared figures and/or tables, authored or reviewed drafts of the paper, and approved the final draft.
- Emilia Peñaherrera-Romero conceived and designed the experiments, performed the experiments, analyzed the data, prepared figures and/or tables, authored or reviewed drafts of the paper, and approved the final draft.

## Field Study Permissions

The following information was supplied relating to field study approvals (i.e. approving body and any reference numbers):

Research permits were issued by Ministerio de Ambiente del Ecuador, 001-16IC-FLO-FAU-DNB/MA, 018-2017-IC-FAU-DNB/MAE, 019-2018-IC-FAU-DNB/MAE, and 006-2015-FAU-DPAP-MA.

## Data Availability

Raw data are available as a Supplemental File.

## Supplemental Information

Supplemental information for this article can be found online at http://dx.doi.org/10.7717/peerj.10461#supplemental-information.

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
