# Peer review of "Invasion history of Harmonia axyridis (Pallas, 1773) (Coleoptera: Coccinellidae) in Ecuador"

_PeerJ, doi:10.7717/peerj.10461_

## Round 0.1 · original submission · Minor Revisions

Both reviewers suggest that you manuscript would be acceptable with some relatively minor revisions. Please follow the recommendations of the reviewers, but especially consider the request for more depth in the discussion (from reviewer 1).

·

Basic reporting

No comment - everything is included in section 4

Experimental design

No comment - everything is included in section 4

Validity of the findings

No comment - everything is included in section 4

Additional comments

The paper presents interesting and apparently thorough information on the spread of a key invasive species in a country previously lacking clear information on the subject. It is well written and there is good use of literature. Whilst I recommend it for publication, the paper is very descriptive and feel that the Discussion lacks depth. With some additions as below I think it would be a much more valuable contribution to the extensive literature on this species. Although I have suggested quite a few changes, I think that most of them should be relatively easy to make. There are also quite a few minor (but rather careless) errors in referencing etc.

The Introduction is clear and concise, but a few changes are suggested:
Line 32 and check elsewhere: citations not in chronological order
L42 ‘…the Harlequin Ladybird’
L45 ‘control agent’
L45 the ‘becoming invasive’ statement should be after the ‘establishing’ and ‘expanding’ statements (i.e. re-order sentence)
L54 the Koch 2003 reference (cited elsewhere) is a useful addition to the statement about prey diversity
L62 there are many primary citations that could be used in addition to the Lucas 2012 book chapter, and some examples could be added – e.g. Ware & Majerus 2008 BioControl 53; Katsanis et al 2013 BioControl 58
L62 delete ‘a’ (‘exerting strong’)
L69 ‘In North and South America’ would be clearer
L69, 75 and check elsewhere: ‘Harmonia’ should be abbreviated
L72 ‘…of the country with H. axyridis…’
L74 delete ‘about’
L79 Were the surveys targeted just at H. axyridis, all ladybirds, or broader than that?
L121 clarify that you mean (I assume) 9 spots on each elytron, thus 19 overall.

Whilst the museum / literature searching look thorough, there is some detail missing about the field sampling in Materials & Methods. It appears from the results that only four direct field observations of the species were made. This is not a problem in itself, but there is no indication of the scope of the field sampling that led to these four records. Since the field sampling is mentioned first in this section it looks like a major part of the study, but the results suggest that this isn’t the case, so perhaps relegate this to the end, i.e. change the order of the various methods. If it was a substantial part of the study but the species was not found in most of the localities searched, this should be made clear.
When (range of months and years) and by whom were the surveys carried out? How were localities selected? (Presumably not random, so what criteria were used to select potential sites, e.g. based on particular habitat characteristics?) Indicate how many sites were searched and at how many was the species NOT found. This would provide greater context on how ubiquitous the species is.

Results
L133 ‘find’ (not ‘found’)
Third paragraph: could a table be constructed summarising the number of records found in each habitat type?
Are all the records of adults? Presumably some are larval / pupal records but I can’t see this indicated.
L145-6 this sentence is method
L148 ‘were’ not ‘are’
L149-151 it would be interesting to add the plant families for those unfamiliar with these species
L156 use past tense
L157 genus in full at start of sentence
L162-8 interesting but does this relate only to the local field sampling, or are some of the noted species associations from literature sources?

Discussion
There are some structural problems with the contents of this section:
The first paragraph either needs to be moved to Introduction, or put more in the context of this study, i.e. start with a reflection on the Ecuador results and then add the other information on Harmonia in the Americas to put the local situation in a wider context. At the moment this feels like the wrong way around. Splitting this long paragraph into one on South America and one on North would also improve it.
Really the second paragraph merely reports some example results and should therefore be moved to that section; or else add some interpretation, e.g. by comparing the habitats mentioned to those reported from other countries, etc.
So the third paragraph of the Discussion is what I would regard as the first part of the report that is proper discussion.
As alluded to earlier, it would be interesting to hear more on the habitats used by Harmonia in Ecuador, and a reflection on this here would be nice. It would be particularly good to hear a little more on the high altitude habitats where the ladybirds were found, e.g. what plant species they were on, etc, at such high altitudes.
L200 ‘All previously published…’?
L212 ‘record’ (not ‘recorded’)
L215 ‘in Ecuador’ duplicated
L218-9 Is this sentence referring to H. convergens or H. axyridis?
L224 The mention of spread rate in the UK comes from Brown et al 2008 BioControl 53 (listed incorrectly as a 2007 paper in the References)
L227-8 clumsy wording that should be tidied up
L237 Similar to above, the Poutsma 2007 paper I believe should be cited as 2008. (The official published version of the BioControl 53 Special Edition was 2008. Check elsewhere for this issue.)
L238 ‘habitats’ (plural)
L245 ‘arthropod’
L247-8 Can any example effects specific to the region be mentioned here?
L249-50 Like above, the statement about impacts on fruit is very general. Can this be put in the context of Ecuador? Is commercial fruit production important (and in regions where Harmonia is common?) and which main fruits are grown?
L250 Start a new paragraph after the part about fruit. Can you add something on how many endemic species are known in Ecuador, thus adding some conservation context?

References
Please thoroughly check this section, as there seem to be quite a few formatting inconsistencies, e.g. doi or DOI? Weblinks / DOIs sometimes given, sometimes not; missing volume and page numbers (e.g. L302)
References by the same first author do not seem to be in sensible order - e.g. see Koch; Brown. Please check elsewhere.
L298 ‘Brown PMJ’ (not ‘MJP’)
L420 ‘Hodek’ mis-spelt

Fig 1
Some extra labelling would be helpful on the map, e.g. some major cities added, or regions labelled. For Discussion: Is there any explanation why nearly all of the records are in the central band of the country, when seemingly the lower altitude areas to both E and W look superficially more suitable (to someone with little knowledge of the geography of Ecuador). This is mentioned in passing (L235) with brief discussion of Amazonian lowlands, but are these on both sides of the Andes? (With apologies if this is an ignorant question!) Are the major urban areas in this central band? A map showing broad habitat types (combined with altitude if appropriate, or shown on a second map) would be nice.

·

Basic reporting

It is a work on the invasion of Harmonia axyridis in Ecuador with relevant results on the introduction and establishment of the species.

Experimental design

The methodology used is adequate.

Validity of the findings

The authors did a good job of collecting data which resulted in an excellent article.

Additional comments

Some suggestions were made in the manuscript itself.

---

## Round 0.2 · accepted · Accept

Thank you for your extensive revisions to your ms which are in line with the directions of the reviewers. I have made some additional edits (see attached), and once these are seen to I think that your ms will be acceptable for publication.